# On the Use of Multivariate Analysis and Land Evaluation for Potential Agricultural Development of the Northwestern Coast of Egypt

**Mohamed El Sayed Said** [1,*], **Abdelraouf. M. Ali** [1,2], **Maurizio Borin** [3,*],
**Sameh Kotb Abd-Elmabod** [4], **Ali A. Aldosari** [5], **Mohamed M. N. Khalil** [6]
**and Mohamed K. Abdel-Fattah** [6]

1   National Authority for Remote Sensing and Space Sciences (NARSS), Cairo 11843, Egypt;
    raouf.shoker@narss.sci.eg
2   Agrarian-Technological Institute of the Peoples Friendship University of Russia, ul. Miklukho-Maklaya 6,
    117198 Moscow, Russia
3   Department of Agronomy, Food, Natural Resources, Animals and Environment—DAFNAE,
    University of Padua, Agripolis Campus, Viale dell' Università 16, 35020 Legnaro (PD), Italy, Veneto
4   Soils & Water Use Department, Agricultural and Biological Research Division, National Research Centre,
    Cairo 12622, Egypt; sk.abd-elmabod@nrc.sci.eg
5   Geography Department, King Saud University, Riyadh 11451, Saudi Arabia; adosari@ksu.edu.sa
6   Soil Science Department, Faculty of Agriculture, ZagazigUniversity, Zagazig 44519, Egypt;
    mmnabil11@gmail.com (M.M.N.K.); mkabdelfattah@zu.edu.eg (M.K.A.-F.)
*   Correspondence: elsayed.salama@narss.sci.eg (M.E.S.S.); maurizio.borin@unipd.it (M.B.);
    Tel.: +20-10-6144-5686 (M.E.S.S.)

**Abstract:** The development of the agricultural sector is considered the backbone of sustainable development in Egypt. While the developing countries of the world face many challenges regarding food security due to rapid population growth and limited agricultural resources, this study aimed to assess the soils of Sidi Barrani and Salloum using multivariate analysis to determine the land capability and crop suitability for potential alternative crop uses, based on using principal component analysis (PCA), agglomerative hierarchical cluster analysis (AHC) and the Almagra model of MicroLEIS. In total, 24 soil profiles were dug, to represent the geomorphic units of the study area, and the soil physicochemical parameters were analyzed in laboratory. The land capability assessment was classified into five significant classes (C1 to C5) based on AHC and PCA analyses. The class C1 represents the highest capable class while C5 is assigned to lowest class. The results indicated that about 7% of the total area was classified as highly capable land (C1), which is area characterized by high concentrations of macronutrients (N, P, K) and low soil salinity value. However, about 52% of the total area was assigned to moderately high class (C2), and 29% was allocated in moderate class (C3), whilst the remaining area (12%) was classified as the low (C4) and not capable (C5) classes, due to soil limitations such as shallow soil depth, high salinity, and increased erosion susceptibility. Moreover, the results of the Almagra soil suitability model for ten crops were described into four suitability classes, while about 37% of the study area was allocated in the highly suitable class (S2) for wheat, olive, alfalfa, sugar beet and fig. Furthermore, 13% of the area was categorized as highly suitable soil (S2) for citrus and peach. On the other hand, about 50% of the total area was assigned to the marginal class (S4) for most of the selected crops. Hence, the use of multivariate analysis, mapping land capability and modeling the soil suitability for diverse crops help the decision makers with regard to potential agricultural development.

**Keywords:** PCA; land capability; crop suitability; GIS; NWCE; Egypt

## 1. Introduction

The world's population will increase to reach approximately 9.7 billion by 2050 [1,2]. The huge population increase will impact agricultural resources as it causes global food security pressure on the lack of agricultural lands [3,4]. Two types of challenges put pressure on governments, namely the increasing population on the one hand, and the decrease in productive land on the other hand [5]. There are two ways that governments can counteract overpopulation: the first is to encourage farmers to increase crop yields by using land fertilizers, pesticides, etc., which affect environmental quality, and the second is to rely on imports to fill the food gap [3,5]. Therefore, it is required to increase the efforts to improve living standards to provide safe food to feed citizens [6,7]. The expansion of new agricultural lands is the goal of developing countries such as Egypt where, the annual growing population rate is 84%, while the strategic crops production such as wheat is insufficient, therefore, the government relies on importation from abroad [8–13]. Moreover, agricultural expansion in arable lands aims to achieve sustainable agricultural development, which depends on the integration of land and water resources and the surrounding environmental factors [14,15]. Agriculture lands in Egypt are confined to the Nile Valley and the Delta that represents approximately 4% of the total area of Egypt [16]. The agricultural sector contributes to 14.5% of the gross domestic product (GDP) of Egypt, whilst representing about 30% of the provision of foreign currency as a result of the profits of exporting agricultural products abroad, and it also contributes to reducing unemployment by 41% [17]. The concept of land assessment belongs to the rate of land performance and its capacity for crop production where the capacity of land depends on the climate and location/geography, the inherent soil characteristics (physical, chemical) and also includes the soil potential for agricultural production [18,19]. The importance of land evaluation helps in selecting the suitable crops based on the soil characteristics and assists the decision makers [15,20]. The excess of salt concentration in soil may damage soil structure by decreasing soil aeration and its permeability, and consequently adversely affects the agricultural production. However, appropriate soil management leads to decreased salt concentration, a decline in sodium percentage, and improved drainage conditions [21,22]. There are several agriculture management practices for saline–sodic and calcareous soils in arid and semi-arid regions, such as improving the soil's physiochemical characteristics by adding organic matter, reducing soil salinity by fresh water leaching, and reducing sodium saturation and the alkaline pH using gypsum and sulfur applications [14]. The soil limitation factors for crop suitability differ from one place to another in Egypt, while the dominant limiting factors in the north of the Nile Delta are the soil salinity, poor drainage and compaction [23–28]. Meanwhile, the hardpan layers, shallow depth of the soil profile, and the rock outcrops as well as the steep slope are the most common limiting factors in the desert lands [15,29,30].

The northwestern coast region is vulnerable to land degradation and desertification processes which lead to reduced soil fertility and cause environmental impacts, as the lower areas in the north suffer from rising salinization and alkalinization, meanwhile the valleys are susceptible to runoff and soil erosion, where surface runoff reached 200 mm/year, and the soil erosion of soil has reached $60\,\mathrm{t\,h^{-1}\,y^{-1}}$ [30]. Over the past five decades, several models for soil capability classification have been proposed to classify the soils according to their chemical and physical properties [18]. De la Rosa et al. [31] suggested micro land evaluation system (MicroLEIS) to test the soil suitability: this system integrates soil characteristics, topography, vegetation cover, land use and climate conditions. Other methods were developed to classify the soil capability for crop production according to their soil profile description and soil characteristics such as slope, texture, salinity and other factors such as the drainage conditions, where each class takes an average value between 0 and 100, where 100 reflects the best conditions and vice versa [32]. The assessment of land capability depends on an evaluation of the soil quality and expresses a capacity of the soil to function in an ecosystem in order to sustain the soil productivity of a crop in parallel with reducing the soil degradation processes, in addition to its ability to perform a number of basic functions such as supporting crop production [24,30], whereas the soil is a complex mixture of organic, inorganic materials and it is influenced by the surrounding

factors such as climate, topography and human activities. The integration of soil characteristics, climatic data, remotely sensed data, water analysis and socio-economic data with GIS modeling assist to develop spatial decision support system (SDSS) for soil management [12,33,34]. Due to the obscure nature of the soil system, it is not easy to evaluate the soil by integrating their properties together. Therefore, multivariate analysis is an appropriate tool for evaluating soil capability zones due to its efficiency in modeling and systematically using vague and imprecise situations [7,13,35]. Principal component analysis (PCA) is considered the most popular model to analyze the physical, chemical, socio–economic and other factors in a multistage analysis to develop an indicator that represents the evaluation of land capability. The PCA and fuzzy clustering means (FCM) methods were used for land suitability evaluation for oil palm and soil quality based on soil characteristics and climate data [36,37]. FCM is an unsupervised clustering method for the data components that enables investigating the accumulation of multiple elements. Clustering can extract the homogenous regions based on different phenomena. Moreover, given the gradually changing nature of soil behaviors, it seems that the fuzzy clustering method can interpret the spatial variation of studied phenomena better than the other methods [38]. The PCA was used to decrease the variables to increase the accuracy of FCM land quality evaluation. Coinciding with the availability of computers, satellite data and GIS in the past two decades have led to improved methods of land assessment, where satellite images provided important information on the status of plants, topography and climate [39,40]. In addition, agglomerative hierarchical clustering (AHC) was used to define the distances between points where the similar points are forming one cluster, then finally all clusters are presented in a dendrogram form [41]. The Northern coast of Egypt depends on seasonal rains during the winters, and the citizens' lack of awareness of suitable crops and water irrigation quantity suit those conditions [30]. Therefore, the purpose of this study is to use multivariate analysis to assess the soil of the study area to determine the most appropriate land use.

## 2. Materials and Methods

### 2.1. Study Area

The investigated area was allocated in Matrouh governorat, northwestern Egypt, close to the Libyan border. The area includes the Sidi Barrani and Salloum districts (Figure 1). It is located between 25°10′ to 26°55′ East and 31°00′ to 31°37′ North. The area covers approximately 918,000 hectares. The study area is characterized by an arid and semi-arid climate, where the average temperature reaches 18 °C in the whole year; except in June, July and August, the temperature ranges between 25 and 30 °C. The annual rainfall fluctuates between 100 and 200 mm/year.

The elevation ranged between 0 and 250 m above sea level in the south, whereas the low elevation values were observed in the parts close to the Mediterranean Sea and the high values were noticed in the plateau that occupies the southern parts. The north of the study area is characterized by a gentle slope, while the south is characterized by a moderate to steep slope (Figure 2). The micro-relief varies from almost flat to undulating with scattered escarpments, and a flat coastal zone about 1–3 km wide. Some principle wadis dissect the escarpment, especially southwest of Sidi Barrani. Wadi is an Arabic term which traditionally refers to a valley that is located in low land and receives a high amount of rainfall compared with the surrounding land forms [42].

The vegetation cover of the study area is changing during the winter and spring seasons as the rainfall is active, whereas the natural vegetation spreads throughout the study area, especially in the wadis and streams. The natural vegetation spreads on the fine sand stacks that keep rain water during the winter season and leads to natural vegetation growth during the autumn and summer seasons [30].

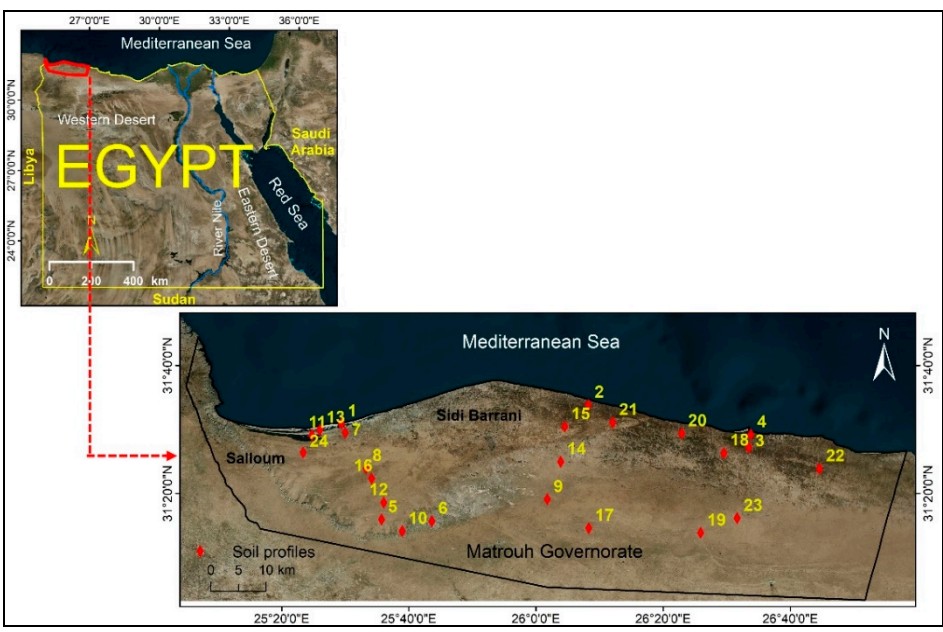

**Figure 1.** Location of the study area in northwestern Egypt and the soil profiles with a red dot.

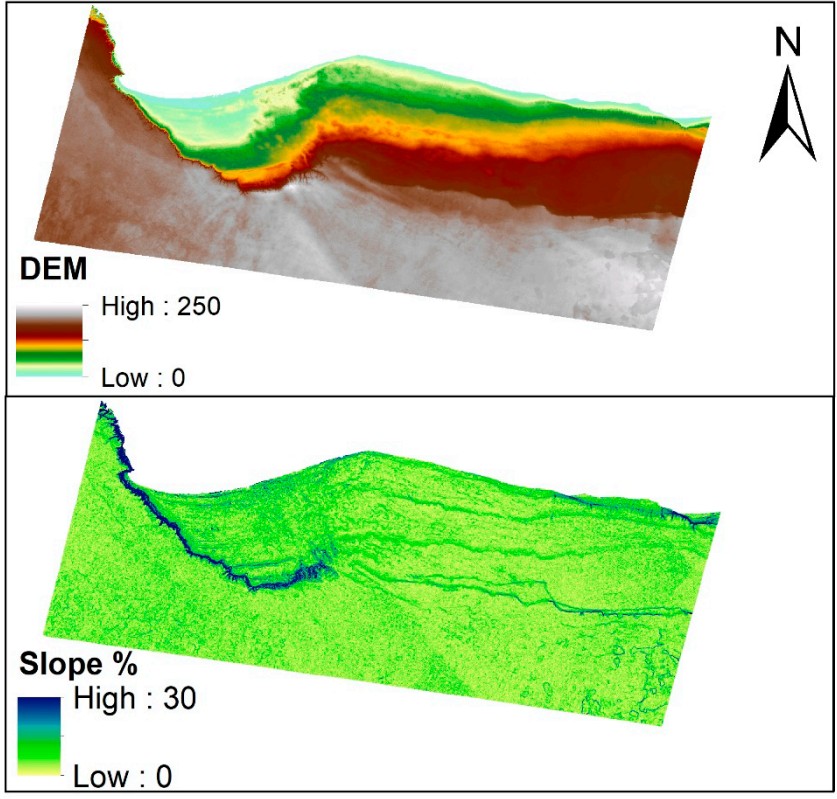

**Figure 2.** Digital elevation model (DEM) in meters and slope (%) of the study area.

## 2.2. Remote Sensing and Spatial Analysis

Remote sensing data were used to link the soil capability with land use and geomorphological units of the study areas, whereas the Sentinel 2 image was acquired 1 July 2020 with spatial resolution 10 m for the red, blue and green bands and 20 m for the near infrared band. On the other hand, the digital elevation model (DEM) with 30 m resolution was used to describe the variation in the surface characteristics and its relation with geomorphological units and soil capability.

The slope percentage in the study area reached 30% in the adjacent escarpments of the plateau while low slope values were observed in the wadis (Figure 2). Sentinel 2 image data, DEM and the field survey were integrated to enhance the visibility of the geomorphological map that was produced by [22] using the method described by [43]. This relied on topographic information such as the slope, aspect, and stream networks of the study area that were extracted using DEM. Consequently, a number of 15 geomorphological units were identified to represent the diverse geomorphological features. All geomorphological units were verified based on a field survey using GPS. Then, the produced geomorphological map was used a base map, where each geomorphic unit was homogeneous in its natural characteristics. Generally, the biophysical characteristics were spatially analyzed [44].

### 2.3. Field Survey and Soil Laboratory Analysis

The field survey relies on the identified geomorphic units of the study area, whereas twenty-four soil profiles were dug to a depth of 150 cm or less according to the presence of hardpan layers, in order to represent geomorphic units (Figure 1). The soil profile samples were determined based on a sample area method that was suggested by [45] where the sample area should be distributed on the lines that pass the geomorphological units without bias. Therefore, some units have more than one profile and another unit has only one profile.

In addition, the units that have large areas have more representative soil profiles compared with the small unit. The descriptions of soil profiles were done in the field using [19]. Soil samples were collected from different layers of soil profiles. Then, the soil physiochemical parameters were analyzed. Table 1 showed the mean soil characteristics of the soil profiles. The following chemical soil properties were determined: salinity (electric conductivity) [46] and soil acidity (pH) in saturated paste [47], cation exchange capacity (CEC, [48]), soil organic matter content by the acid-dichromate potassium and titration method [49]. In addition, the particle size distribution and macronutrients were measured according to [40]. Soil nitrogen (N) was determined using the Kjeldahl method [50]. The soil phosphorus was determined using the spectrophotometer device according to [51] (P) and potassium (K) was determined using a flame photometer. The available soil potassium contents were measured using flame photometry [50]. The exchangeable sodium percentage (ESP) was determined using methods of [52,53]. The soil classification of each soil profile was done according to [54].

### 2.4. Statistical Analysis

PCA is a statistical procedure that uses orthogonal transformation to convert a set of observations of possibly correlated variables (entities each of which takes on various numerical values) into a set of values of linearly uncorrelated variables called principal components (PCs). PCs having eigenvalues greater than one were retained whereas PCs less than 1 were subtracted away. Soil properties were summarized using PCA.

Before performing the principal component analysis (PCA), a linear relationship between the soil variables were checked using the Pearson correlation coefficient. This analysis requests sampling adequacy, therefore, the Kaiser–Meyer–Olkin (KMO) was done to measure the sampling adequacy for the overall data set. If the KMO value is greater than 0.50, the PCA would be suitable. In addition, the Bartlett's test was performed, and if the *p* value of Bartlett's test value is less than 0.05, this indicates that the PCA may be suitable for the work [55,56]. Generally, all the statistical analyses were performed using SPSS software version 25.

Based on the PCA results, the soil profiles which were considered as objects for soil capability evaluation were classified into dissimilar clusters using the AHC methods according to each location characterized by a group of soil variables (chemical, physical, biological). Through this analysis, dissimilar groups of soil variables were arranged together graphically in a structure called a dendrogram of dissimilarity.

**Table 1.** The variation of soil characteristics for each geomorphic unit of the study area.

| Profile No. | Depth, cm | pH | ESP, % | OM, % | EC, dS/m | CaCO₃, % | CEC cmolc/kg | Texture | Drainage | Rock Fragment, % | AN, mg kg⁻¹ | AP, mg kg⁻¹ | AK, mg kg⁻¹ | Erodibility, ton/ha/Year |
|---|---|---|---|---|---|---|---|---|---|---|---|---|---|---|
| 1 | 120 | 8.5 | 4.41 | 0.64 | 10.03 | 32.55 | 23.5 | LS | WD | 6.4 | 11.4 | 10 | 29 | 0.18 |
| 2 | 115 | 8.32 | 4.42 | 0.64 | 2.88 | 25.86 | 20.5 | SCL | WD | 7.3 | 13 | 10 | 29 | 0.23 |
| 3 | 110 | 8.6 | 4.45 | 0.22 | 3.01 | 57.75 | 14.6 | SL | WD | 7.4 | 5.38 | 3.7 | 9.7 | 0.03 |
| 4 | 35 | 8.59 | 1.00 | 0.48 | 2.57 | 55.35 | 5.6 | LS | PD | 4.6 | 10.4 | 7.8 | 22 | 0.02 |
| 5 | 150 | 8.28 | 2.95 | 0.14 | 1.47 | 40.86 | 5.6 | S | WD | 6.5 | 3.69 | 2.6 | 6.3 | 0.06 |
| 6 | 60 | 8.2 | 2.06 | 0.24 | 2.09 | 12.57 | 9.9 | S | ID | 3.6 | 5.82 | 3.9 | 11 | 0.07 |
| 7 | 120 | 7.8 | 1.65 | 0.33 | 16.67 | 4.60 | 9.0 | SL | ID | 6.8 | 8.21 | 5.1 | 15 | 0.02 |
| 8 | 55 | 7.7 | 8.31 | 0.54 | 5.96 | 5.95 | 6.3 | SL | PD | 4.1 | 11.6 | 8.6 | 25 | 0.04 |
| 9 | 45 | 8.27 | 3.52 | 0.34 | 3.07 | 51.73 | 11.3 | S | PD | 5.3 | 8.92 | 5.4 | 14 | 0.10 |
| 10 | 35 | 7.6 | 1.25 | 0.23 | 2.68 | 8.03 | 5.8 | S. | PD | 6.2 | 5.2 | 3.7 | 11 | 0.19 |
| 11 | 150 | 8.24 | 3.99 | 0.50 | 42.99 | 48.62 | 13.9 | SL | WD | 8.3 | 12.4 | 7.9 | 21 | 0.01 |
| 12 | 85 | 8.3 | 2.31 | 0.37 | 5.35 | 51.57 | 11.3 | SL | ID | 6.2 | 8.93 | 6 | 17 | 0.02 |
| 13 | 115 | 8.2 | 2.52 | 0.42 | 96.56 | 23.39 | 10.7 | SL | WD | 4.2 | 10.4 | 6.7 | 19 | 0.04 |
| 14 | 95 | 7.9 | 3.10 | 0.50 | 16.50 | 7.80 | 12.1 | LS | WD | 3.8 | 8.1 | 7.2 | 20 | 0.11 |
| 15 | 120 | 8.41 | 1.18 | 0.19 | 3.91 | 69.50 | 6.8 | S | ID | 7.4 | 5.64 | 3.1 | 9.1 | 0.06 |
| 16 | 90 | 7.59 | 1.10 | 0.30 | 3.01 | 20.70 | 5.4 | S | WD | 5.5 | 6.86 | 4.7 | 14 | 0.07 |
| 17 | 20 | 7.82 | 1.18 | 0.38 | 4.58 | 10.80 | 5.6 | S | PD | 4.8 | 9.44 | 6.3 | 17 | 0.03 |
| 18 | 30 | 7.95 | 2.40 | 0.38 | 4.80 | 9.80 | 6.2 | S | PD | 4.7 | 15 | 7.2 | 14 | 0.10 |
| 19 | 55 | 8.2 | 1.20 | 0.20 | 9.50 | 8.40 | 8.1 | SL | ID | 3.9 | 9.2 | 8.8 | 16 | 0.08 |
| 20 | 120 | 8.13 | 0.95 | 0.40 | 2.54 | 25.67 | 9.2 | S | WD | 5.9 | 10.3 | 6.7 | 18 | 0.21 |
| 21 | 115 | 8.13 | 1.13 | 0.28 | 1.91 | 45.97 | 6.2 | S | WD | 6.8 | 7.62 | 4.6 | 13 | 0.15 |
| 22 | 80 | 8.08 | 0.90 | 0.53 | 2.16 | 38.86 | 4.2 | LS | ID | 4.3 | 11.7 | 8.5 | 24 | 0.10 |
| 23 | 150 | 8.19 | 0.74 | 0.18 | 2.71 | 57.51 | 5.0 | S | WD | 6.7 | 4.68 | 3 | 8.3 | 0.15 |
| 24 | 120 | 7.88 | 0.90 | 0.20 | 3.18 | 25.00 | 10.1 | S | WD | 5.3 | 8.4 | 3.5 | 9.2 | 0.23 |

ESP, exchangeable sodium percentage; OM, organic matter; EC; electric conductivity; CEC, cation exchange capacity; Drainage: WD, well drained; PD, poorly drained; ID, imperfectly drained. Texture: L, loamy; SCL, sandy clay loam; SL, sandy loam; LS, loamy sand; S, sandy. AN, available nitrogen; AP, available phosphorous; AK, available potassium.

*2.5. Land Evaluation*

The assessment of soil capability depends on determining the relationship of soil proprieties and agriculture suitability. In this context, the multivariate analysis aims to classify soil capability based on the harmony of the soil properties in each class. Therefore, determining the capability of soil was based on the integration between PCA and AHC methods, as the PCA shows a visual representation of the dominant patterns in order to identify the similarities and differences among soil properties [57]. The assessment output was grouped into five broad classes: C1—a highly capable land, C2—moderately high class, C3—moderate class, C4—low and C5—marginal.

The Almagra model of MicroLEIS [58,59] is a qualitative approach characterized by being built on using soil factors and the favorable conditions for each crop. The soil factors considered in the model are: profile depth (p), texture (t), carbonate (c), salinity (s), drainage (d), and sodium saturation (a). The model was used to assess the soil suitability for ten traditional crops, annuals, e.g., wheat, sunflower, soybean, maize, sugar beet, potato; semiannual, e.g., alfalfa and perennials, e.g., citrus, olive and peach. Almagra defines the soil suitability qualitatively through five classes: S1—optimum, S2—high, S3—moderate, S4—marginal and S5—not suitable. Almagra was calibrated previously in several studies within the Mediterranean, semi-arid, and arid regions [14,60,61]. Figure 3 illustrates the flowchart methodology of soil evaluation based on the integration of soil information remote sensing data and GIS using multivariate analysis as following.

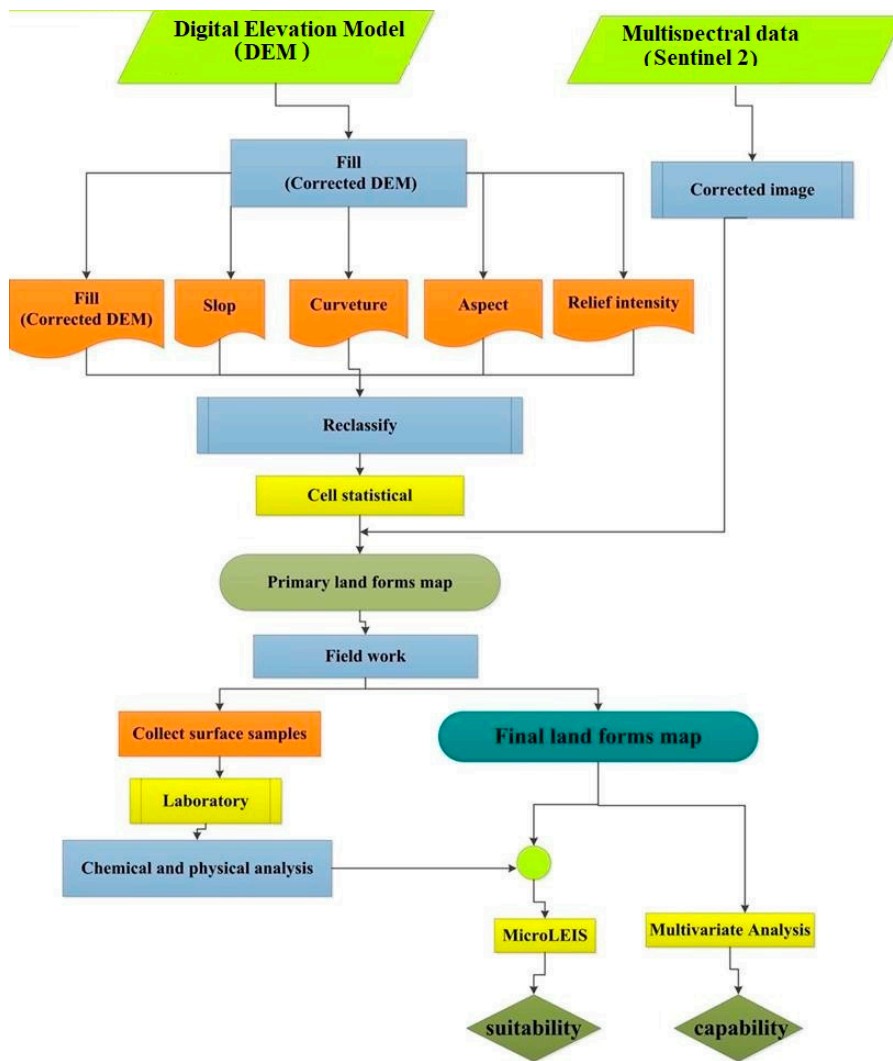

**Figure 3.** Flowchart of the land evaluation of the study area.

## 3. Results

### 3.1. Geomorphological Map and Soil Characteristics

Fifteen geomorphological units were recognized and modified using the integration of DEM, sentenal2 images and ground truth data. Figure 4 and Table 2 show the fifteen geomorphic units that describe the variations of the study area. The landscape of the study area was characterized by an almost flat to undulating surface, while the ridges that were located at the north were intersected by short and long wadis.

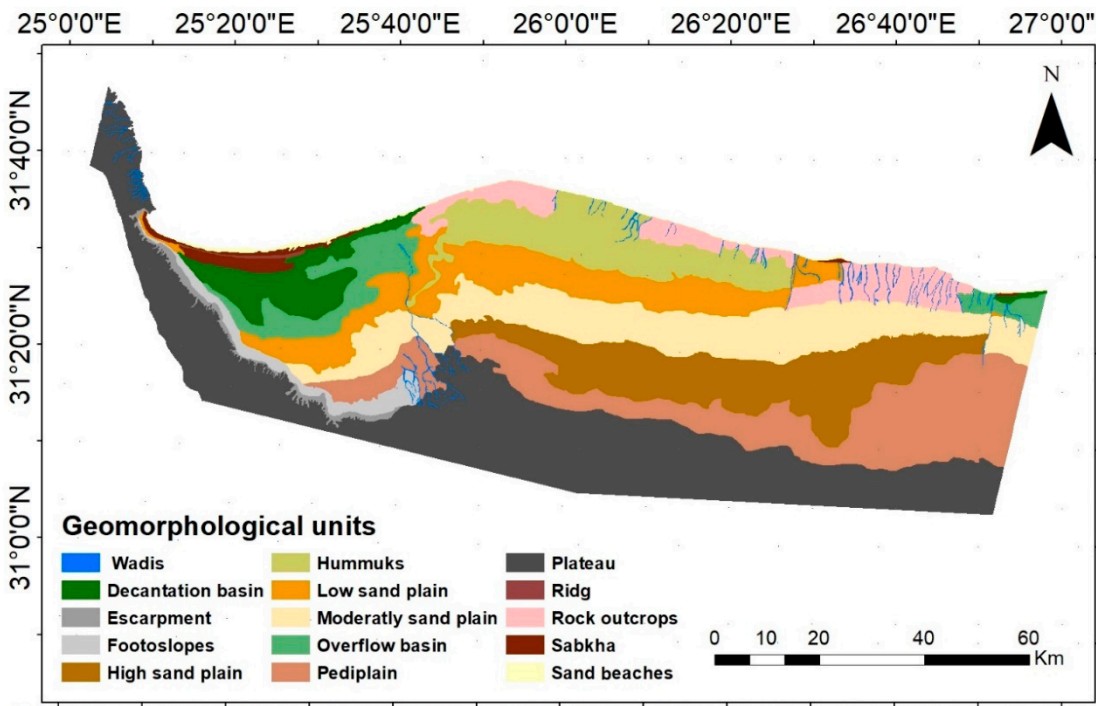

**Figure 4.** Geomorphological map of the study area. Source: modified after Mohamed et al. [30].

**Table 2.** The areas of geomorphological units.

| Geomorphologic Units | Area, ha | Area, % |
|---|---|---|
| Wadis | 7201.39 | 0.93 |
| Decantation basin | 28,015.05 | 3.62 |
| Escarpment | 6905.64 | 0.89 |
| Foot slopes | 13,015.67 | 1.68 |
| High sand plain | 86,488.80 | 11.18 |
| Hummuks | 57,821.26 | 7.47 |
| Low sand plain | 71,629.91 | 9.26 |
| Moderatly sand plain | 101,777.62 | 13.16 |
| Overflow basin | 31,048.51 | 4.015 |
| Pediplain | 108,714.35 | 14.05 |
| Plateau | 209,502.84 | 27.09 |
| Ridg | 1091.26 | 0.14 |
| Rock outcrops | 36,830.13 | 4.76 |
| Sabkha | 8095.55 | 1.04 |
| Sand beaches | 5095.41 | 0.65 |
| **Total** | **773,233.46** | **100** |

The following lines describe the dominant geomorphic units of the study area

**Wadis**—this unit is widespread in the northwestern coast of Egypt and represents one of the most diagnosed geomorphological units. It is located in gentle slopes and receives high amounts of runoff compared with the surrounding upland. Wadis of the study area include different lithological material, and their sedimentary structures vary widely from gravel to mud. Generally, the soils of wadis are considered the most suitable for agriculture development in the northwestern coast. The dominants soil textures of the Wadis were sand clay loam, sandy loam and sand. The soil depth ranges between 110 and 150 cm based on their position in the valley and slope. The low values of soil salinity were observed where it varied between 1.5 and 3.0 dS/m. The soil pH fluctuated between 8.3 and 8.6. The organic matter (OM) content is very low where it ranges between 0.14 and 0.64%. The carbonate content ($CaCO_3$) is high (25–40%). The CEC depends upon the clay and organic matter contents, while it ranged from 5.6 to 20 cmolc kg$^{-1}$. Meanwhile, the soil classification of wadis was determined to be *Typic Haplocalcids*.

**Basins**—basin units are described as low lands where rainfall and drained water accumulate into the outlet. Therefore, basins include the accumulative surface runoff, and nearby streams that run downslope towards the shared outlet. This unit is divided into decantation basins and overflow basins:

**I-Decantation basins are** characterized by deep and moderate soil depth, which ranges between 100 and 120 cm. The soil texture fluctuates between sandy loam and loamy sand. The soil salinity is slightly moderate to high, varying between 3 and 16 dS/m. The OM is low as it ranges between 0.2 and 0.3%. The soil pH is categorized as moderately alkaline and ranges between 7.8 and 7.9. The carbonate content is slightly moderate to high when it ranges between 4.5 and 25%. The CEC ranged between 9.0 and 10.1 meq/100 g, and the soils are classified as *Typic Torripsamments*.

**II-Overflow basin**—this unit is the upper areas of basins. The soil texture ranges between sand and loamy sand. Soil salinity fluctuates from low to moderate, where it reached to 3 dS/m. The OM is very low where it is less than 0.3%. The $CaCO_3$ content varies from moderate to high where it reached 20%. CEC was low (5 cmol/kg). Soils are classified as *Typic Torrifluvents*.

**Foot slopes** are mainly located in the marginal land as plateau areas, were the surface is covered by weathered fragments. The soil depth is shallow as a result of the presence of hardpan layers that are observed at depths ranged between 35 and 55cm. Coarse sand and sandy loam are the dominant soil textures, $CaCO_3$ content is slightly moderate, ranging between 5.95 and 8%. The electric conductivity (EC) is slightly low as it ranges between 2.6 and 5.9 dS/m, and the pH is slightly alkaline (7.6 to 7.7). The OM content is relatively low (0.23 and 0.54). The CEC varied between 5.6 and 6.6 cmol kg$^{-1}$, while the soil was classified as *Lithic Torriorthents*.

**Sand plains** occupy an area of about 33.6% of the total area, classified into **I—high, II—moderate and III—low sand plains**. Soil textures were varied between sandy, loamy sand and sandy loam. Soil profile depth ranged between 80 and 150 cm. The EC value fluctuated from low to high (2.1 to 16.5 dS/m). The Soil pH was moderately alkaline except for some parts of the unit, and it ranged between 7.9 and 8.3. The OM content is very low to low where it ranged between 0.18 and 0.53%. The $CaCO_3$ content is varied in a wide range from moderate to high (7.8–57.51%). The CEC was low where it ranged between 4.2 and 12.1 cmol kg$^{-1}$, unlike the soils classified as *Typic Torripsamments* and *Typic Torrifluvents*.

**Hummuks** areas are characterized by the undulating surface and it has an accumulation of sand dunes due to the active wind, the units of which occupy an area of about 7% of the total area. The dominated soil textures varied between sandy and loamy sand. The soil depths were moderately deep and deep (115–150 cm). The EC was low as it ranged between 1.9 and 3.9 dS/m. Soil pH is mostly moderate alkaline (8.13–8.4). The OM content is very low (0.19 and 0.40%). The $CaCO_3$ content is high (45–69%). The CEC was low where it ranged between 4.2 and 9 cmol kg$^{-1}$, while the soil was classified as *Typic Torripsamments*.

**Pediplain** unit occupies 14% of the total area, gently undulating and almost featureless in their surface, which was formed by the erosion processes over a long time. This unit has a shallow soil depth (20–60 cm). Soil is slightly saline where the EC values ranged between 2.09 and 9.5 dS/m. Soil pH is

moderately alkaline (7.8–8.2). The OM content was very low (0.2–0.38%). The CaCO$_3$ content ranged between 8.4 and 10.8%. The CEC was low where it ranged between 5.6 and 9.9 cmol kg$^{-1}$, whereas the soils were classified as *Lithic Torripsamments*.

**Rock outcrops**—this unit was a gently undulating surface with rock fragments with a diameter of a few centimeters to a few meters. Soil depths were shallow and ranged between 30 and 45 cm. The EC ranged between 2.57 and 4.8 dS/m, and the soil pH fluctuated between 7.95 and 8.59. The OM content was low where it ranged between 0.34 and 0.48%. The CaCO$_3$ content ranged between 9.80 and 55.35%. The CEC ranged between 5.6 and 11.3 cmol kg$^{-1}$, while the soils of this unit were classified as *Lithic Torripsamments*.

**Sabkha**—this unit was located in the low land at the north of the study area, and the soil texture of this unit was loamy sand. The soil salinity was varied between high and very high (42–96 dS/m). The OM was low (0.4 to 0.5%). The CaCO$_3$ content was high and it ranged between 23 and 48%. The CEC was varied from 10.7 to 13.9 cmol kg$^{-1}$, and the soil was classified as *Typic Haplosalids*.

**Sand beaches**—this unit is a strip of sand close to the Mediterranean Sea, and the soil texture of this unit was loamy sand. The soil salinity reached 10 dS/m. The OM was low as it reached 0.64%. The CaCO$_3$ content was high and it reached 32.5%. The CEC was 23.5 cmol/kg. The soil was classified as *Typic Torripsamments*.

The study area has others geomorphological units that were not considered for the assessment of the soil capability and crop suitability such as Ridges (narrow strip of the hardpan layer of calcium carbonate that resisted erosion) and the plateau as it has a very shallow soil depth.

### 3.2. Multivariate Statistical Analysis

The Pearson correlation analysis illustrates the correlations between the soil variables and among them as shown in Table 3. There is a positive significant correlation between the pH and CaCO$_3$ where ($r$ = 0.72). Moreover, there was a positive significant correlation between the ESP and CEC ($r$ = 0.88) and with available K ($r$ = 0.46). While there was a logical positive correlation between the organic matter and CEC ($r$ = 0.44), available N ($r$ = 0.89), available P ($r$ = 0.80) and available K ($r$ = 0.78). Moreover, there was a correlation between the CEC and the available P ($r$ = 0.54) and the available K ($r$ = 0.62).

**Table 3.** Correlation matrix (Pearson).

| Variables | Depth | pH | ESP | OM | EC | CaCO$_3$ | CEC | Rock fr. | AN | AP | AK | Er. |
|---|---|---|---|---|---|---|---|---|---|---|---|---|
| pH | 0.28 | | | | | | | | | | | |
| ESP | 0.03 | 0.08 | | | | | | | | | | |
| OM | −0.23 | 0.01 | **0.41** | | | | | | | | | |
| EC | 0.25 | 0.05 | 0.12 | 0.15 | | | | | | | | |
| CaCO$_3$ | 0.38 | **0.72** | −0.10 | −0.25 | −0.09 | | | | | | | |
| CEC | 0.13 | 0.15 | **0.88** | **0.44** | 0.12 | −0.13 | | | | | | |
| Rock fr.% | **0.62** | 0.30 | 0.05 | −0.17 | −0.08 | **0.55** | 0.10 | | | | | |
| Av. N | −0.25 | 0.00 | 0.31 | **0.89** | 0.23 | −0.25 | 0.39 | −0.19 | | | | |
| Av. P | −0.21 | 0.12 | 0.40 | **0.80** | 0.18 | −0.27 | **0.54** | −0.24 | **0.84** | | | |
| Av. K | −0.11 | 0.10 | **0.46** | **0.78** | 0.22 | −0.19 | **0.62** | −0.14 | **0.76** | **0.94** | | |
| Er. | 0.19 | −0.11 | −0.16 | 0.07 | −0.29 | −0.13 | 0.11 | 0.10 | 0.05 | 0.06 | 0.07 | |

ESP, exchangeable sodium percentage; OM, organic matter (%); EC; electric conductivity (dSm$^{-1}$); CEC, cation exchange capacity (cmolc kg$^{-1}$); Rock fr., rock fragments; AN, available nitrogen; AP, available phosphorous; AK, available potassium; Er., erodibility (ton/ha/year). Note: values in bold are different from 0 with a significance level alpha = 0.05.

Table 4 shows the factor loadings and component score coefficient outputs that illustrate the higher factor loads using the varimax method. The most representative physical and chemical indicator for PC1, as it is closely correlated with the ESP, OM, CEC, N, P and K, while the second factor (PC2) was correlated with soil depth, pH, CaCO$_3$, and rock fragment, whereas the third (PC3) was correlated with K, the fourth factor (PC4) contributes with the first in ESP and the fifth (PC5) was correlated with EC.

**Table 4.** Summarization of the principal component analysis (PCA) analysis.

| PCs parameters | PC1 | PC2 | PC3 | PC4 | PC5 | | | | |
|---|---|---|---|---|---|---|---|---|---|
| Eigenvalue | 4.43 | 2.45 | 1.35 | 1.23 | 1.06 | | | | |
| Variability, % | 36.9 | 20.42 | 11.27 | 10.28 | 8.85 | | | | |
| Cumulative, % | 36.9 | 57.32 | 68.59 | 78.86 | 87.72 | | | | |
| | **Factor loadings** | | | | | **Component Score Coefficient** | | | | |
| | PC1 | PC2 | PC3 | PC4 | PC5 | PC1 | PC2 | PC3 | PC4 | PC5 |
| Depth cm | −0.25 | 0.70 | 0.20 | 0.25 | 0.48 | −0.06 | 0.29 | 0.15 | −0.19 | 0.45 |
| pH | −0.01 | 0.73 | −0.28 | −0.41 | −0.19 | 0.00 | 0.30 | −0.22 | 0.33 | −0.18 |
| ESP,% | 0.61 | 0.34 | 0.02 | 0.55 | −0.41 | 0.14 | 0.14 | 0.02 | −0.45 | −0.38 |
| OM, % | 0.88 | 0.00 | 0.02 | −0.24 | 0.07 | 0.20 | 0.00 | 0.01 | 0.19 | 0.07 |
| EC, dS/m | 0.24 | 0.14 | −0.56 | 0.34 | 0.65 | 0.06 | 0.06 | −0.41 | −0.28 | 0.61 |
| $CaCO_3$, % | −0.38 | 0.74 | −0.21 | −0.39 | −0.16 | −0.09 | 0.3 | −0.15 | 0.31 | −0.15 |
| CEC, cmolc $kg^{-1}$ | 0.69 | 0.41 | 0.22 | 0.44 | −0.25 | 0.16 | 0.17 | 0.17 | −0.35 | −0.24 |
| Rock fr.,% | −0.28 | 0.74 | 0.26 | 0.02 | 0.10 | −0.06 | 0.30 | 0.20 | −0.01 | 0.09 |
| AN, mg $kg^{-1}$ | 0.87 | −0.03 | −0.05 | −0.29 | 0.18 | 0.20 | −0.01 | −0.04 | 0.23 | 0.16 |
| AP, mg $kg^{-1}$ | 0.93 | 0.04 | −0.01 | −0.23 | 0.06 | 0.21 | 0.02 | −0.01 | 0.18 | 0.05 |
| AK, mg $kg^{-1}$ | 0.91 | 0.15 | 0.02 | −0.14 | 0.07 | 0.21 | 0.06 | 0.02 | 0.11 | 0.07 |
| Er., ton/ha/year | 0.03 | −0.01 | 0.87 | −0.20 | 0.26 | 0.01 | 0.00 | 0.64 | 0.18 | 0.24 |

ESP, exchangeable sodium percentage; OM, organic matter; EC; electric conductivity; CEC, cation exchange capacity; AN, available nitrogen; AP, available phosphorous; AK, available potassium.

Figure 5 shows the correlations between the different variables based on the angle changes between the vectors, where the degree of the angle expresses the correlations between the variables and each other. The results showed the linkage between some variables by the low angle that means there is a high correlation in the positive direction such as $CaCO_3$ with OM, while the variables were linked together by angle around 90°, and in this case there was no correlation between the variables; on the other hand, there were some variables linked by angle close to 180°, which indicates a correlation in the negative direction. Bartlett's sphericity and KMO test (Table 5) show suitable p values as ($p < 0.05$) of the acceptable level. Furthermore, Bartlett's sphericity test showed that the p value was lower than 0.001. According to the Bartlett's sphericity and KMO test, the PCA was applicable to the current data.

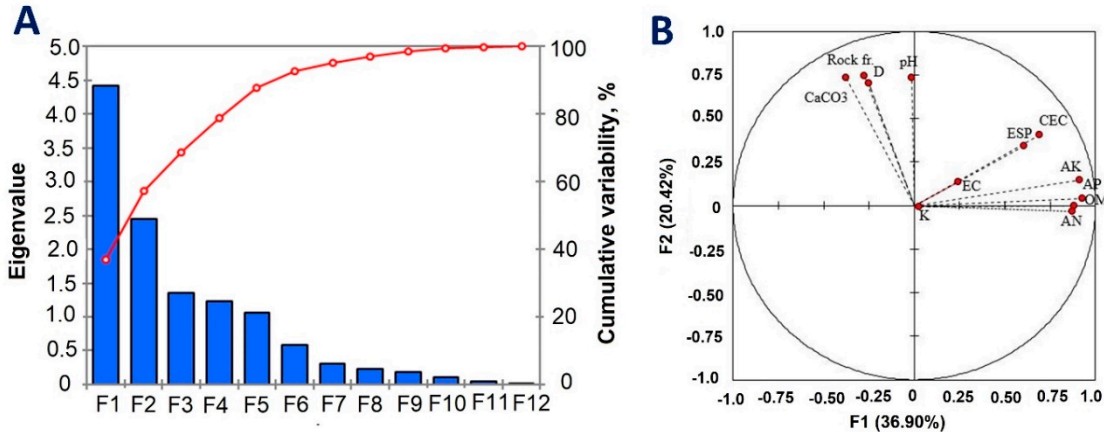

**Figure 5.** The scree plot of eigenvalue (**A**) and principal component analysis bi-plot (**B**) for soil properties.

**Table 5.** Bartlett's sphericity and the Kaiser–Meyer–Olkin (KMO) test.

| Kaiser–Meyer–Olkin measure of sampling adequacy: | 0.52 |
|---|---|
| Bartlett's sphericity test: | |
| Chi-square (Observed value) | 206.90 |
| Chi-square (Critical value) | 85.96 |
| Degree of freedom (DF) | 66.00 |
| *p*-value | <0.0001 |
| Alpha | 0.05 |

### 3.3. Land Capability Based on PCA

The PCA results were used to evaluate the soil capability taking in consideration the variation of topography and climate conditions, therefore the geomorphological units of the study area were used as a base map for the land capability evaluation as each geomorphic unit has specific characteristics such as elevation, slope, aspect and geomorphic features. PCA was performed using the soil physicochemical properties of 24 soil profiles. PCA was applied for the soil characteristics based on the eigenvalues, proportions of variance and cumulative factors by the PCs. Table 4 showed the soil indicator groups. Furthermore, the PCs that had eigenvalues > 1 were retained whereas the PCs < 1 were neglected. Therefore, the first five groups were selected as their eigenvalues were bigger than 1. Table 4 and Figure 5 show these five PCs, which explain the cumulative variance of 87.72% of the studied variables, as the first component explains about 36.9%, the second 20.52%, third 11.27%, fourth 10.28% and the fifth 8.89% of the total variance. Figure 6 illustrates the hierarchical dendrograms for the classification of the soil properties, where each of the five clusters was represented by soil profiles that contain a set of similar soil properties.

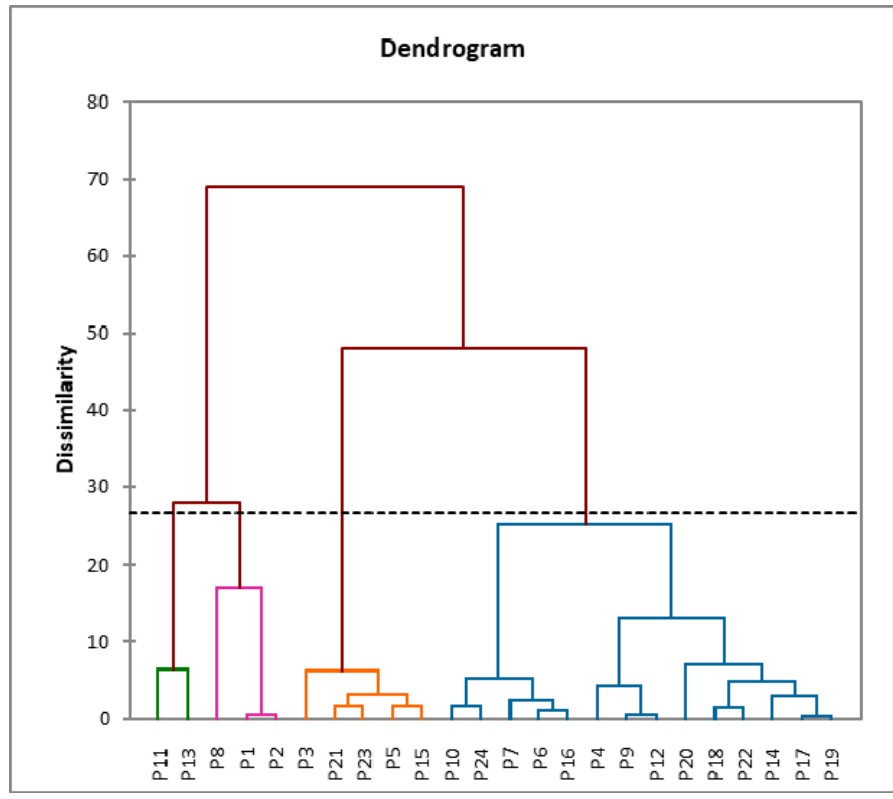

**Figure 6.** Dendrogram for agglomerative hierarchical clustering.

The land capability map of the study area was produced based on the results of the PCA as the map reflects the five groups previously obtained as shown in Figure 7. Table 6 illustrates the main statistical analysis of the soil properties for soil callability classes (C1 to C5) where;

Highly capable class (C1) represents 7% of the total area, a class characterized by deep soil profiles (<115 cm), a high concentration of macronutrients (N, P, K) and low salinity values. On the other hand, the high contents of calcium carbonates (21.45%) were considered as a limiting factor and the soil of this class is vulnerable to erosion where the erodibility factor reached 0.15 ton/ha/year.

Moderately high class (C2) occupies 52% of the total area, the soils of this class were characterized by a deep soil profile and low soil salinity. The limiting factors were the low contents of macronutrients and high contents of calcium carbonates ($\approx$54%).

Moderate class (C3) occupies 29% of the total area. The soil chemical analysis of this class showed low values of salinity, CEC and OM, while the limiting factors of this unit were high contents of calcium carbonates and the shallow soil depth (56 cm).

Low class (C4) represents 11% of the total study area. This unit has a certain number of limitations, such as the soil depth and very low contents of macronutrients. In addition, the soils of this class are vulnerable to soil erosion where, the erodibility factor reached 0.11 ton/ha/year.

Marginal class (C5) occupies a small area (0.9%) and is characterized by the deep soil and high pH values, while the high content of salts (69 dS/m) represents the major limiting factor for soil capability.

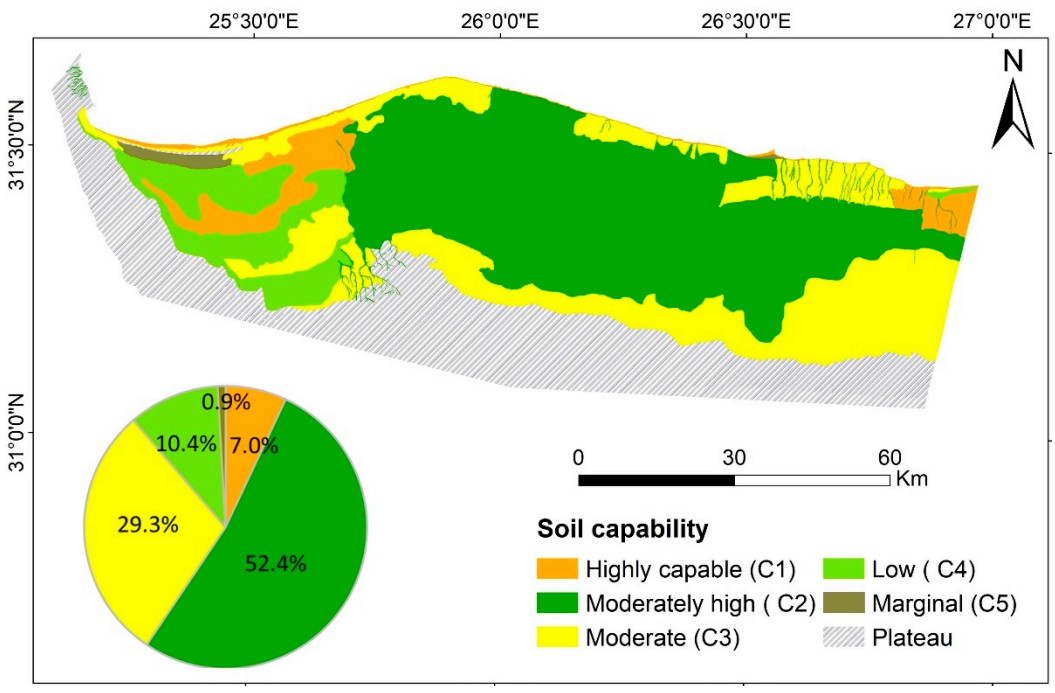

**Figure 7.** Land capability units and the pie chart represent the percentage area.

### 3.4. Soil Suitability

Soil profiles were evaluated based on their suitability for crop production taking into consideration the crop requirements, whilst each crop has a specific requirement for soil properties to achieve the maximum yield production. Therefore, the application of the Almagra model is based on the use of soil characteristics such as profile depth (p), texture (t), carbonate (c), salinity (s), drainage (d), and sodium saturation (a).

The assessment results showed that the soil suitability for the studied crops was varied from S2 to S5 with different limiting factors in each class based on the geomorphological unit. The suitability assessment was examined for ten horticultural and field crops (i.e., wheat, maize, olive, alfalfa, potato, sugar beet, peach, citrus, soybean and fig), Figure 8.

**Table 6.** The main statistical characteristics of the soil properties in five clusters.

| Classes | Depth, cm | pH | ESP, % | OM, % | EC, dSm$^{-1}$ | CaCO$_3$, % | CEC, cmolc kg$^{-1}$ | Rock Fragment, % | AN, mg kg$^{-1}$ | AP, mg kg$^{-1}$ | AK, mg kg$^{-1}$ | Erodibility, ton/ha/Year | Limiting Factors |
|---|---|---|---|---|---|---|---|---|---|---|---|---|---|
| 1 | 118.3 ± 2. | 8.2 ± 0.42 | 5.7 ± 2.25 | 0.6 ± 0.06 | 6.3 ± 3.59 | 21.45 ± 13.84 | 23.4 ± 2.9 | 5.9 ± 1.65 | 11.98 ± 0.85 | 9.75 ± 0.96 | 27.44 ± 2.23 | 0.15 ± 0.1 | C, Er |
| 2 | 129 ± 19.49 | 8.3 ± 0.19 | 2.1 ± 1.57 | 0.2 ± 0.05 | 2.6 ± 0.96 | 54.3 ± 11.22 | 7.6 ± 3.95 | 6.96 ± 0.42 | 5.4 ± 1.45 | 3.4 ± 0.77 | 9.2 ± 2.33 | 0.08 ± 0.06 | C, Fer |
| 3 | 56. 7 ± 25.98 | 8.13 ± 0.24 | 1.9 ± 1.01 | 0.44 ± 0.17 | 5.7 ± 4.64 | 28.9 ± 20.62 | 8.2 ± 2.94 | 4.8 ± 0.83 | 10.2 ± 2.07 | 7.1 ± 1.12 | 17.96 ± 3.36 | 0.08 ± 0.06 | D, Ca |
| 4 | 90 ± 30 | 7.8 ± 0.25 | 1.4 ± 0.46 | 0.25 ± 0.05 | 5.5 ± 6.25 | 12.2 ± 6.32 | 8. ± 2.28 | 5.48 ± 1.21 | 6.9 ± 1.42 | 4.2 ± 0.7 | 11.86 ± 2.22 | 0.11 ± 0.09 | D, Er, Fer |
| 5 | 132.5 ± 24.75 | 8.2 ± 0.03 | 3.3 ± 1.04 | 0.45 ± 0.05 | 69.8 ± 37.7 | 36 ± 17.84 | 12.3 ± 2.20 | 6.25 ± 2.9 | 11.4 ± 1.46 | 7.3 ± 0.78 | 20.35 ± 1.58 | 0.02 ± 0.2 | S, C |

ESP, exchangeable sodium percentage; OM, organic matter; EC, electric conductivity; CEC, cation exchange capacity; AN, available nitrogen; AP, available phosphorous; AK, available potassium. Limiting factors: C, calcium carbonate; Er, erodibility; Fer, fertility; D, depth; S, salinity.

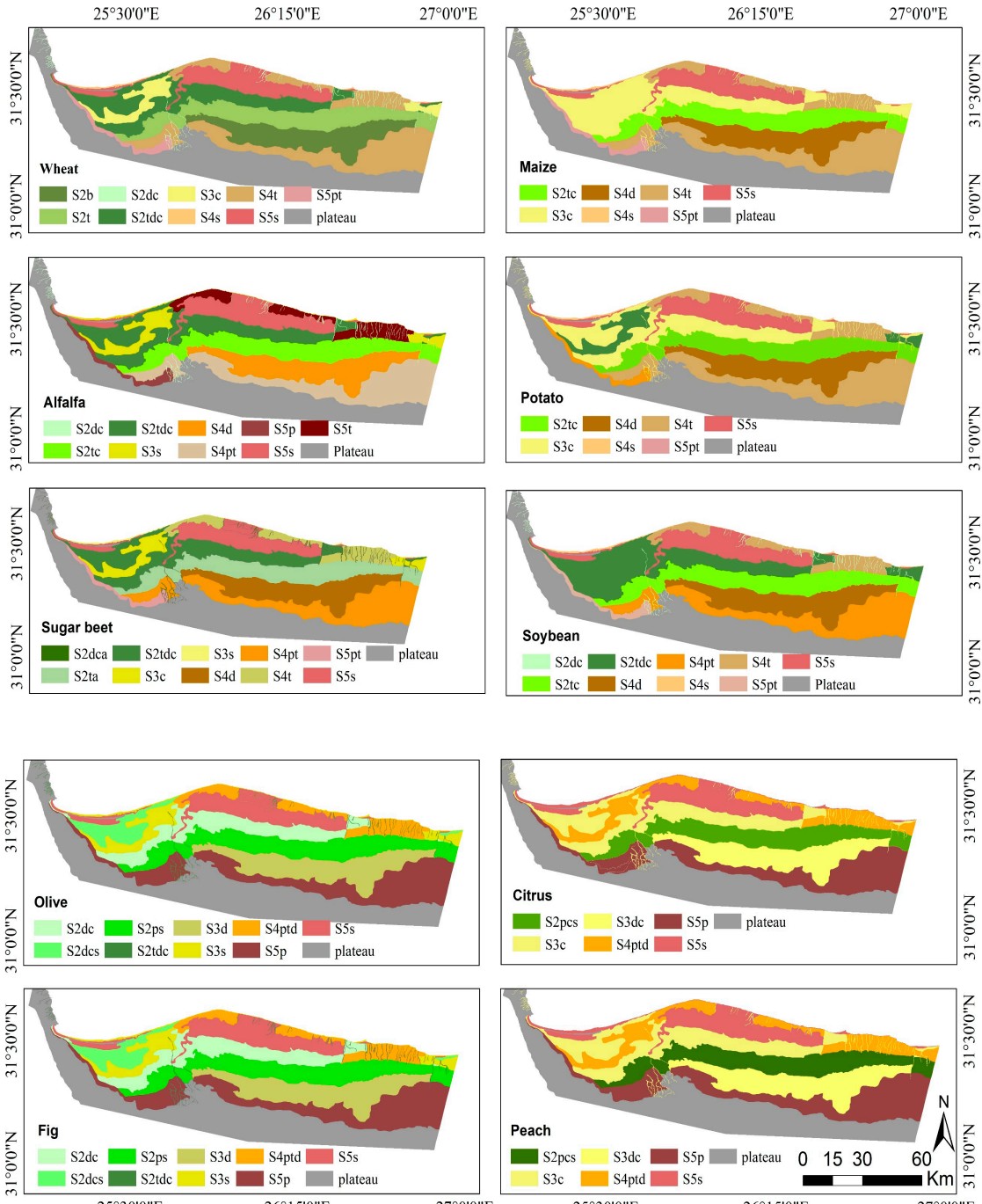

**Figure 8.** Crop suitability map for some field and horticultural crops. The main limiting factors; s, salinity; t, texture; a, sodium saturation; d, drainage; c, carbonate content; p, profile depth.

The geomorphic units such as the plateau, escarpment and ridge were not considered in the suitability evaluation, which represents 28% of the total study area. The results indicate that 37% of the total area was highly suitable for olive, fig, wheat, alfalfa, sugar beet, soybean crops. Furthermore, 18% of the area was highly suitable (S2) for citrus and peach. An area of about 37% of the arable cultivated area was moderately suitable (S3) for the following crops: wheat, olive, alfalfa, sugar beet and fig, but the limiting factors differed in each crop, while 50% of the total area was classified as S4 for most crops. On the other hand, the results showed that about 56% of the study area has marginal suitability (S4) for both peach and citrus crops.

Generally, the area has good potential for cultivating field crops in order to achieve sustainable agricultural development, but water for irrigation is highly needed.

## 4. Discussion

### 4.1. Geomorphology and Soils

Using Sentinel 2 images leads to increased clarification details of landscape where the high and low lands, basin and wadis boundaries were identified [30,62,63]. The soils that are located in southern parts of the study area were classified as *Lithic Torripsamments*, whereas the soil profiles' depth is very shallow and this is considered as a main obstacle to soil capability and crop suitability [64,65]. On the contrary, the soil types *Typic Torripsaments*, *Typic Torrifluvents* and *Typic Haplocalcids* were allocated in the wadi, decantation basins and sand plain geomorphic units with deep soil profile and low soil salinity [66]. Generally, the organic matter content in the study area is very low due to the lack of agricultural activity and monsoon rains, which is consistant with [67].

### 4.2. Multivariate Statistical for Land Evaluation

There is no doubt that due to the similarity of soil properties, grouping them into similar clusters is not easy, as it depends on understanding whether each soil characteristic is increasing or decreasing. For example, the increase in salt concentration leads to negative impacts on crop productivity, contrary to the increases in soil nutrient concentration and organic matter which aid to improve soil fertility. Hence, it is necessary to classify soil properties and link the classes with soil capability and crop suitability. Consequently, the multivariate statistical analysis is suitable to classify multiple variables of soils [68]. The results reflected that there are positive correlations between soil characteristics such as pH and $CaCO_3$ (r = 0.72) or ESP and CEC (r = 0.88). These results are logical, as the soil pH is affected by increasing the percentage of calcium carbonate of soils [69]. The obtained factor loading illustrates the acceptable grouping of soil properties and confirm the ability of PCA for group soil properties in different clusters [70]. The first PCs showed an accumulation of 36.9% of the soil characteristics where the ESP, OM, CEC, N, P and K were allocated in these PCs due to the association of the natural conditions and soil formation processes in the study area [71,72]. The second PCs showed a grouping of 20% of the data of the soil depth, pH, $CaCO_3$, and rock fragments; this cluster deals with the natural environment of the areas that have a shallow soil depth have also a high percentage of rock fragment. In addition, this cluster has a high concentration of pH and $CaCO_3$%.

The highly capable class (C1) represents the soils that have good physical and chemical characteristics, in addition to a low erosion vulnerability, Furthermore, the soils of this class were located in the areas with active agricultural management [73]. The soils of the moderately high class (C2) occupy most of the study area and are characterized by low soil fertility due to the lack of agricultural usage. As the study area was located in arid climate conditions with neglected rainfall in the whole year except in the winter season, cultivation is limited by the winter season and the availability of irrigation water. During this cultivation season, farmers may use fertilizers in order to enhance crop growth, also activating microorganisms. Therefore, the lack of farming usage during the whole year may cause low soil fertility [74]. The main limiting factors of the moderate class (C3) and low class (C4) were the soil depth, high percentage of calcium carbonates, and hardpan layers [75,76], yet C4 has more limitations compared to C3. The soils of the marginal class (C5) occupied the low land (sabkhas) whereas the high salt content was due to the sea water percolation and high evaporation [77]. Therefore, the soils of C5 cannot be used for agriculture consistently, as the agriculture management process is difficult.

### 4.3. Crop Suitability

The assessment of soil suitability for crops was performed using the soil characteristics and crop requirements [78]. About 38% of the study area was classified as S2 for the following crops:

wheat, alfalfa, sugar beet and soybean, which were located in the geomorphic units of wadis, decantation basin, high sand plain, moderate sand plain, and low sand plain. The soil characteristics meet the requirements of those crops, and this is consistent with [79–83]. On the other hand, the rest of the study area are varied in its suitability for wheat crop, between S3 and S5 [63,82]. The highly suitable class (S2) for the maize and potato crops was assigned only to the moderate sand plain unit, while the rest of the study area ranged between S3 and S5 based on the variation of soil limitations and climatic conditions [63,83]. The degree of the soil suitability of the olive and figs crops were classified as S2 in soils of wadis, decantation basin, high sand plain, moderate sand plain, low sand plain, and the overflow basin.

The limiting factors were the soil depth, soil texture, and climate [81]. About 13% of the soils of the study area were classified as S2 for peach and citrus crops, 25% as S3 and the rest of the area was classified as S4 and S5, due to the many limitations that reduce the suitability for both peaches and citrus crops such as soil profile depth and soil texture [84]. Generally, the results indicated that the optimum soil suitability class (S1) was not observed for any crop, as always there is one or more soil characteristics considered as a limiting factor, which is in agreement with [66]. Moreover, the northwest coast is one of the desert areas in Egypt suffering from land degradation which leads to decreased agricultural production. The assessment of soil capability, crop suitability, remote sensing analysis and GIS helps the decision making for sustainable agriculture development [85–90].

## 5. Conclusions

Multivariate analysis classification techniques can deal with various soil variables. The principal component analysis (PCA) and agglomerative hierarchical cluster analysis aid to classify soil capability based on the correlations and interactions between soil proprieties. The PCA classified the soil capability into five classes, which differed according to the number of soil limitations. The main limiting factors for soil capability in the southern parts of the study area was the shallow soil depth where hardpan layers were observed in the subsurface, while the high salinity and alkalinity represent the major limitations in the low elevated land that located in north of the study area. Whilst the area is vulnerable to the wind and water erosion processes, the crop suitability varied between high and not suitable classes for wheat, maize, fig, potato, and citrus. However, the soils that were located in the wadis, sandy plains and basins geomorphic units were of a highly suitable class for most of the field and horticultural crops.

The use of multivariate analysis, soil capability and crop suitability based on soil physiochemical properties assist in understanding the soil function and to assess the soils under different conditions. Likewise, remote sensing data contribute to map the geomorphic unites that were used as a base map for soil assessment. GIS techniques were considered as a main tool to spatialize the variations of soil capability and crop suitability in order to achieve the optimal land use planning in such new reclaimed areas.

**Author Contributions:** All the authors substantially contributed to this article M.K.A.-F. and A.M.A. conceptualized the study and developed the methodology. The satellite imagery was analyzed by M.E.S.S., M.K.A.-F. and M.K.A.-F. accomplished the data analysis and wrote a draft of the manuscript. A.A.A., S.K.A.-E., M.M.N.K. and M.B. contributed to reviewing and editing the manuscript. All authors have read and agreed to the published version of the manuscript.

**Funding:** The Research was supported by University of Padova, DOR project.

**Acknowledgments:** The authors would like to thank the National Authority for Remote Sensing and Space Sciences for supporting the satellite data and image processing. In addition, we thank, Faculty of Agriculture, Zagazig University for the laboratory analysis. The authors would like to extend the thank to National Research Center, Egypt for its support. The authors would like to extend their sincere appreciation to the Deanship of Scientific Research at King Saud University, through the Research Group Project no. RGP-VPP-275 for support. We would like to extend our gratitude to the Research Group Project no 5-100, RUDN University, Russia, for their support. Finally, we would like to thank the Department of Agronomy, Food, Natural Resources, Animals and Environment, DOR project, University of Padova, for the guidance and the progress of the work.

**Conflicts of Interest:** The authors would like to hereby certify that there was no conflict of interest in the data collection, processing the data, the writing of the manuscript, and the decision to publish the results.

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
