# Peer review of "On the Use of Multivariate Analysis and Land Evaluation for Potential Agricultural Development of the Northwestern Coast of Egypt"

_agronomy, doi:10.3390/agronomy10091318_

Round 1
Reviewer 1 Report
Manuscript «On the Use of Multivariate Analysis and Land Evaluation for Potential Agricultural Development of the Northwestern Coast of Egypt» is devoted to the application of various methods and approaches to assess the soil resources of certain territories.
There are several scenarios for the growth of the human population on the planet, I think it would be correct to indicate the relative which scenario such a value is presented.
Line 54. [1,2] (.
Extra parenthesis
Line 69 [20; 15] Should be separated by comma
Line 72 [21,22]. Another font for links. The same errors are found further in the text.
A cross-cutting error throughout the text, confusion with indents of new paragraphs, somewhere they are, and of different sizes, somewhere not.
Remove breaks between paragraphs.
Confusion in pagination.
The description of Figure 1 is very limited. At least there is not enough designation for red dots.
Line 166 the name of the subsection is written in a different font and a different size in comparison with the main text.
2.3. Field survey and soil laboratory analysis
Table 1 somewhere indicates sandy, and somewhere S. It is necessary to unify.
The line spacing in the text is not uniform, it is necessary to unify.
Line 28. The parenthesis on links is not closed.
Figure 3 caption differs in font size. The name of the picture must begin with a capital letter.
Table 3. The correlation of potassium to potassium is quite obvious, it makes sense to remove this value.
233 Judging by the rest of the text, there should not be a break for a new paragraph.
Figure 4 needs to be centered.
Figure 4. The title of the picture, different font sizes are used in the context of one title.
I would suggest that the authors describe in more detail the resulting clusters and how they relate to theoretical data.
Figure 7. Name with a capital letter.
Line 395 An extra break in the text.
It makes sense to format table 6 differently, for example, as table 4 (orientate horizontally) and place it on a separate page.
There are a lot of minor flaws in the text, it is necessary to carefully consider the formatting of the manuscript and bring it to the proper state, in accordance with the template
Author Response
There are several scenarios for the growth of the human population on the planet, I think it would be correct to indicate the relative which scenario such a value is presented.
Line 54. [1,2] (.
Extra parenthesis
We have removed the extra parenthesis, line 45
Line 69 [20; 15] Should be separated by comma
We have use 1comma, line 68
Line 72 [21,22]. Another font for links. The same errors are found further in the text.
We have unified the font
A cross-cutting error throughout the text, confusion with indents of new paragraphs, somewhere they are, and of different sizes, somewhere not.
Remove breaks between paragraphs.
We have removed the break between the paragraphs
Confusion in pagination.
The description of Figure 1 is very limited. At least there is not enough designation for red dots.
We have referred to the red dots in the caption of Figure 1, also we referred to this figure in the part of 2.3. Field survey and soil laboratory analysis, in order to have more description about this figure.
Line 166 the name of the subsection is written in a different font and a different size in comparison with the main text.
2.3. Field survey and soil laboratory analysis
We have modified the text size and unified the font
Table 1 somewhere indicates sandy, and somewhere S. It is necessary to unify.
We have referred to the soil texture class by only letters and in the table footnote we insert the explanation of each letter
The line spacing in the text is not uniform, it is necessary to unify.
Line 28. The parenthesis on links is not closed.
We have checked and modified
Figure 3 caption differs in font size. The name of the picture must begin with a capital letter.
We have checked and modified
Table 3. The correlation of potassium to potassium is quite obvious, it makes sense to remove this value.
We agree and removed the column, also we modified this letter to be Er. that refer to the erodibility.
233 Judging by the rest of the text, there should not be a break for a new paragraph.
We have deleted the break
Figure 4 needs to be centered.
We have centered
Figure 4. The title of the picture, different font sizes are used in the context of one title.
We have unified the font
I would suggest that the authors describe in more detail the resulting clusters and how they relate to theoretical data.
Table 6 and text (line 414 to 433) describe the cluster classes (C1 to C5) and how they relate to theoretical data
Figure 7. Name with a capital letter.
We have corrected
Line 395 An extra break in the text.
The extra break has been deleted
It makes sense to format table 6 differently, for example, as table 4 (orientate horizontally) and place it on a separate page.
We have improved the table and place it on a separated page (landscape orientation)
There are a lot of minor flaws in the text, it is necessary to carefully consider the formatting of the manuscript and bring it to the proper state, in accordance with the template
We have checked carefully the whole document and improved the manuscript format in accordance with the template. Finally, we would like to thank you so much for your comments that helped us to improve our manuscript
Reviewer 2 Report
Line 46: [1,2]( -> [1,2]
Line 48-50: This sentence has no verb.
Line 52: [5,3] -> [3,5]
Line 53: “dramatic growth of population rate”: this is already described. Remove.
Line 54-55: I consider that in most countries, it is difficult to “expand new agricultural lands”. Thus, I do not agree with “the expansion of new agricultural lands is the goal of governments around the world”.
Line 64-67: This should include climate and location/geography.
Line 67-69: This sentence has no verb.
Line 71-72: The authors should describe one or two examples as appropriate soil management.
Line 83-85: Not clear. Rewrite.
Line 85: [31] suggested ***** -> De la Rosa et al. [31] suggested ****
Line 91: itis ???
Line 94: “the soil is a complex mixture of organic compounds and influences”, not clear. Rewrite.
Line 95: [33-35] -> Remove from the beginning of the sentence.
Line 101: analysis -> analyze?
Line 102: [37,38] -> Remove from the beginning of the sentence.
Line 103: FCM is not familiar. Please explain this briefly.
Line 134: “high amount of rainfall when heavy rain occurs” seems not to make sense. Please rewrite.
Line 151: This part is already explained. Please remove either.
Line 167-169: The authors describe 24 profiles were dug to represent geomorphic units. This indicates one profile represent 1 geomorphic unit? Please explain how one profile/site was selected? This is very important. The authors should explain the logics of the representativeness.
Line 177-178: I cannot imagine how P was determined using a flame photometer. Please show the reference and explain the method briefly.
Tables .
% should be g kg-1
CaCO,3 -> CaCO3
Ava N: please explain the definition. What form?
Ava. P, K: please explain the definition. What form?
ppm should be mg kg-1
significant digits should be more carefully considered.
The order of the sections 2.4 and 2.5 should be changed.
Line 228: This part seems to contradict with line 167-169. This indicates that in some geographical units, more than 2 soil profiles were obtained. Please clarify.
Line 469-480: Most part is not relevant.
Line 482-483: this sentence should be removed.
Line 488-489: How “knowledge on the threshold” relates to multivariate statistical analysis? I feel these two sentences are not logically connected.
Line 492-493: grammatically wrong.
Line 505: The authors should describe the reason/mechanism why the lack of agricultural usage causes low soil fertility.
Line 500-515: Most parts are not relevant in Discussion, since the authors do not seem to discuss, just simply describe the definition of each class.
Line 533-534: This sentence has no verb.
Author Response
Line 46: [1,2]( -> [1,2]
We have removed the parentheses
Line 48-50: This sentence has no verb.
We have checked and modified
Line 52: [5,3] -> [3,5]
We have checked and modified
Line 53: “dramatic growth of population rate”: this is already described. Remove.
We have removed this sentence line 52 - 54
Line 54-55: I consider that in most countries, it is difficult to “expand new agricultural lands”. Thus, I do not agree with “the expansion of new agricultural lands is the goal of governments around the world”.
We have deleted “around the world” and focused only on the developing countries such as Egypt, so the text has modified as suggested.
Line 64-67: This should include climate and location/geography.
We have included climate and location/geography in line 68
Line 67-69: This sentence has no verb.
We have added the verb line 69-70
Line 71-72: The authors should describe one or two examples as appropriate soil management.
We have added a paragraph to describe some appropriate soil management practices line 73-76
Line 83-85: Not clear. Rewrite.
We have modified
Line 85: [31] suggested ***** -> De la Rosa et al. [31] suggested ****
We have added
Line 91: itis ???
We have modified
Line 94: “the soil is a complex mixture of organic compounds and influences”, not clear. Rewrite.
We have rewrite
Line 95: [33-35] -> Remove from the beginning of the sentence.
We have removed from the beginning
Line 101: analysis -> analyze?
In this sentence, the word analysis is more suitable than analyze (multivariate analysis)
Line 102: [37,38] -> Remove from the beginning of the sentence.
We have removed from the beginning
Line 103: FCM is not familiar. Please explain this briefly.
We have explained the FCM briefly, lines 108-112
Line 134: “high amount of rainfall when heavy rain occurs” seems not to make sense. Please rewrite.
We have modified
Line 151: This part is already explained. Please remove either.
We have deleted this part
Line 167-169: The authors describe 24 profiles were dug to represent geomorphic units. This indicates one profile represent 1 geomorphic unit? Please explain how one profile/site was selected? This is very important. The authors should explain the logics of the representativeness.
We have inserted a new paragraph to clarify the methodology of selecting soil profiles, lines178-181
Line 177-178: I cannot imagine how P was determined using a flame photometer. Please show the reference and explain the method briefly.
We have corrected, while P was determined by using spectrophotometer, the method was described by Van Reeuwijk, 2002
Tables .
% should be g kg-1
We have considered and modified in tables 1 and 6
CaCO,3 -> CaCO3
We have corrected
Ava N: please explain the definition. What form?
Ava N, available inorganic nitrogen
Ava. P, K: please explain the definition. What form? ;
Ava.P, available inorganic phosphorous; Ava. K, available inorganic potassium
ppm should be mg kg-1
We have changed
Significant digits should be more carefully considered.
We have carefully considered the significant digits
The order of the sections 2.4 and 2.5 should be changed.
We prefer to keep the current order of the sections, as we used the statistical analysis first in evaluating soil capability, then we used Almagra model for evaluating soil suitability for ten traditional crops
Line 228: This part seems to contradict with line 167-169. This indicates that in some geographical units, more than 2 soil profiles were obtained. Please clarify.
Line 228 illustrates the geomorphology map of the study that contains (15 units) while 167-169 explains the number of soil profiles (24 soil profiles) which is greater than the number of geomorphology units as some units contain more than one profiles.
Line 469-480: Most part is not relevant.
We have modified this part
Line 482-483: this sentence should be removed.
We have considered and removed
Line 488-489: How “knowledge on the threshold” relates to multivariate statistical analysis? I feel these two sentences are not logically connected.
We deleted the threshold and modified the text to make it clearer. The evaluation of soils depends on the classification of each soil characteristic and its impact on soil capability and crop suitability, for example soil salinity has different classes and there is a threshold value for each class.
Line 492-493: grammatically wrong.
We have corrected
Line 505: The authors should describe the reason/mechanism why the lack of agricultural usage causes low soil fertility.
As the studied area was located in arid climate conditions that has neglected rainfall in the whole year except in winter season, the cultivation is limited by winter season and the availability of irrigation water. During this cultivated season farmers may use fertilizers in order to enhance the crop growth, also the microorganisms will be activated. Therefore, the lack of farming usage during the whole year it may cause low soil fertility.
Line 500-515: Most parts are not relevant in Discussion, since the authors do not seem to discuss, just simply describe the definition of each class.
We have modified this part
Line 533-534: This sentence has no verb.
We have modified the sentence and make it clearer
Finally, we would like to thank you so much for your time and corrections.
Round 2
Reviewer 1 Report
The authors did a great job of finalizing the article. I hope that my comments helped them in this. Thank you very much for commenting on a number of points that were not clear to me, and possibly for future readers.
In my opinion, an article in this form may be allowed to be considered.
Author Response
Thanks for the reviewing.
I have made the comments.